# DeiSAM: Segment Anything with Deictic Prompting

**Hikaru Shindo**[1], **Manuel Brack**[1,2], **Gopika Sudhakaran**[1,3],
**Devendra Singh Dhami**[3,4], **Patrick Schramowski**[1,2,3,5], **Kristian Kersting** [1,2,3,6]

[1]Technical University of Darmstadt        [2]German Research Center for AI (DFKI)
[3]Hessian Center for AI (hessian.AI)        [4] Eindhoven University of Technology        [5] LAION
[6] Centre for Cognitive Science, Technical University of Darmstadt
{firstname.lastname}@tu-darmstadt.de, d.s.dhami@tue.nl

## Abstract

Large-scale, pre-trained neural networks have demonstrated strong capabilities in various tasks, including zero-shot image segmentation. To identify concrete objects in complex scenes, humans instinctively rely on *deictic* descriptions in natural language, *i.e.* , referring to something depending on the context, *e.g. "The object that is on the desk and behind the cup."*. However, deep learning approaches cannot reliably interpret these deictic representations due to their lack of reasoning capabilities in complex scenarios. To remedy this issue, we propose DeiSAM, which integrates large pre-trained neural networks with differentiable logic reasoners. Given a complex, textual segmentation description, DeiSAM leverages Large Language Models (LLMs) to generate first-order logic rules and performs differentiable forward reasoning on generated scene graphs. Subsequently, DeiSAM segments objects by matching them to the logically inferred image regions. As part of our evaluation, we propose the Deictic Visual Genome (DeiVG) dataset, containing paired visual input and complex, deictic textual prompts. Our empirical results demonstrate that DeiSAM is a substantial improvement over data-driven neural baselines on deictic segmentation tasks.

## Introduction

Recently, large-scale neural networks have achieved substantial advancements in various tasks at the intersection of vision and language. One such challenge is grounded image segmentation, wherein objects within a scene are identified through textual descriptions. For instance, Grounding Dino (Liu et al. 2023b), combined with the Segment Anything Model (Kirillov et al. 2023), excels at this task if provided with appropriate prompts. However, a well-documented limitation of data-driven neural approaches is their lack of reasoning capabilities (Shi et al. 2023; Huang et al. 2023). Consequently, they only perform well for textual prompts directly describing or naming the targets and fail for complex prompts, as demonstrated in Fig. 1.

In contrast, humans identify objects through structured descriptions of complex scenes referring to an object depending on the context, e.g., *"An object that is on the boat and holding an umbrella."*. These descriptions are referred to as *deictic representations* and were introduced to artificial intelligence research motivated by linguistics (Agre and

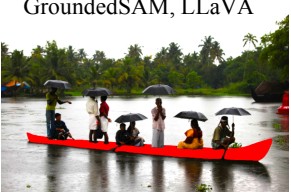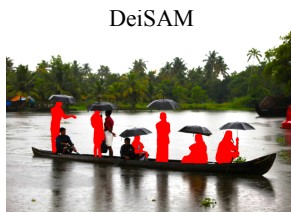

GroudedSAM, LLaVA                    DeiSAM

🤖 📢 *"An object that is on the boat and holding an umbrella."*

Figure 1: **DeiSAM segments objects with deictic prompting.** A segmentation result of DeiSAM (right), and that of GroudedSAM[1]and LLaVA (Chen et al. 2023) (left) given a visual scene and deictic representation as prompt.

Chapman 1987), and subsequently applied in reinforcement learning (Finney et al. 2002). Although deictic representations play a central role in human comprehension of scenes, current approaches fail to interpret them faithfully due to their poor reasoning capabilities.

To remedy these issues, we propose DeiSAM, which combines large-scale neural networks with logic reasoners to segment objects with deictic representations. The DeiSAM pipeline is highly modular and provides a sophisticated integration of large pre-trained networks and neuro-symbolic reasoners. Specifically, we leverage Large Language Models (LLMs) to generate logic rules for a given deictic prompt and perform differentiable forward reasoning (Shindo et al. 2023a,b) with scene graph generators (Zellers et al. 2018). Our reasoner is efficiently combined with neural networks and leverages forward propagation on computational graphs. The result of this reasoning step is used to ground a segmentation model that reliably identifies the objects that best match the input.

Overall, we make the following contributions:

- We propose DeiSAM, a modular, neuro-symbolic reasoning pipeline on LLMs and scene graphs for object segmentation with complex textual prompts.
- We demonstrate *semantic unification*, where similar entities are unified using textual embeddings.

---

[1]https://github.com/IDEA-Research/Grounded-Segment-Anything

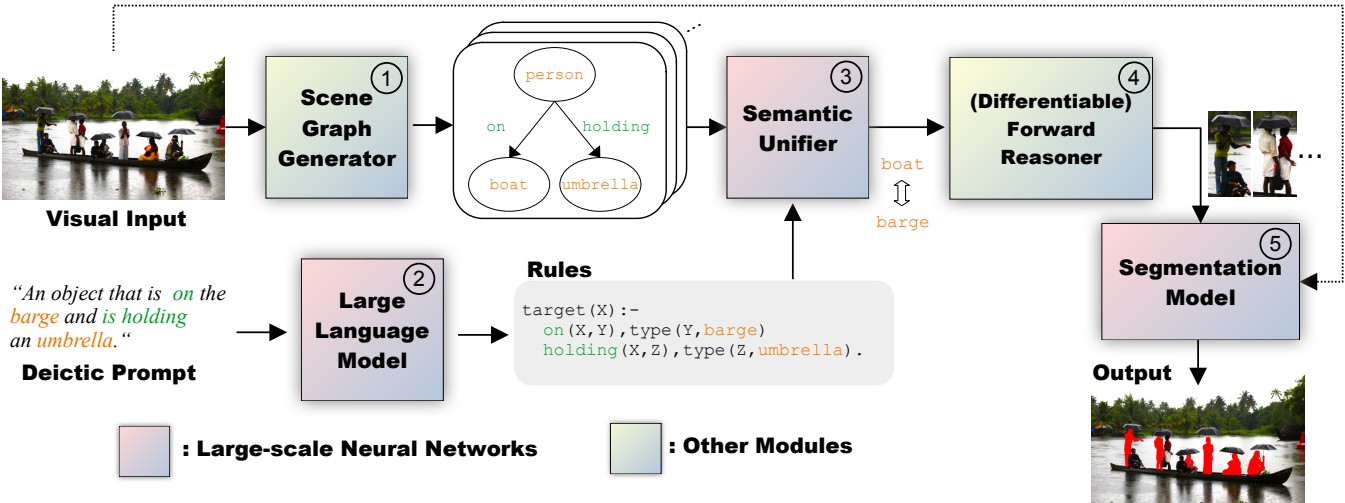

Figure 2: **DeiSAM architecture.** A pair of an image and a deictic prompt is given as input. A scene graph is generated out of the image, and logic rules are generated out of the deictic prompt by a large language model. The generated scene graph and rules are fed to the *Semantic Unifier* module, where similar terms are unified to perform reasoning jointly, *e.g.* `boat` in the scene graph and `barge` in generated rules are regarded as the same term. Forward reasoner infers target objects specified by the textual deictic prompt. To this end, the object segmentation is performed in extracted image crops. Since forward reasoners can be differentiable (Shindo et al. 2023a), gradients could be propagated to learn each module. (Best viewed in color)

- For evaluation, we introduce a novel Deictic Visual Genome (DeiVG) benchmark that contains visual scenes paired with deictic representations, *i.e.*, complex textual identifications of objects in the scene.
- We empirically show that DeiSAM outperforms neural baselines such as GroundedSAM and LLaVA on the proposed task.

## DeiSAM

Let us now devise the DeiSAM pipeline, by giving a brief overview of its modules before describing essential components in more detail. Fig. 2 shows a schematic overview of the proposed workflow.

First, an input image is transferred into a graphical representation using a **(1) scene graph generator**. Specifically, a scene graph comprises a set of triplets $(n_1, e, n_2)$, where entities $n_1$ and $n_2$ have relation $e$. For example, a *person* $(n_1)$ is *holding* $(e)$ an *umbrella* $(n_2)$. Consequently, each triplet can be interpreted as a logic atom, or simply a *fact*, $e(n_1, n_2)$. To perform reasoning on these facts, the paired textual deictic prompt needs to be interpreted as a structured logical expression. For this processing step, DeiSAM leverages **(2) large language models**, which can generate logic rules for deictic descriptions, given sufficiently restrictive prompts. In our example, the LLM would translate the prompt *"An object that is on the barge and holding an umbrella."* to the following rules:

```
target(X):-on(X,Y),type(Y,barge),
        holding(X,Z),type(Z,umbrella).
```

However, users often use terminology different from that of the scene graph generator. For example, *barge* and *boat* target the same concept but will not be trivially matched. To

bridge the semantic gap, we introduce a **(3) semantic unifier**. This module leverages word embeddings of labels, entities, and relations in the generated scene graphs and rules to match synonymous terms by modifying rules accordingly. The semantically unified rules are then compiled to a **(4) forward reasoner**, which computes logical entailment using forward chaining (Shindo et al. 2023a). The reasoner identifies the targeted objects and corresponding bounding boxes from the scene graph. Lastly, we segment the object by feeding the cropped images to a **(5) segmentation model**.

Now, let us investigate the two core modules of DeiSAM in detail. Specifically, we look into how DeiSAM generates logic rules and performs reasoning.

### LLMs as Logic Generators

To perform reasoning on textual deictic prompts, we need to identify corresponding rules. For this translation, we use LLMs to parse textual descriptions to logic rules by using a system prompt like:

```
1  Given a deictic representation and
       available predicates, generate rules
       in the format.
2  The rule's format is either
3  target(X):-pred(X,Y),type(Y,const).
4  or
5  target(X):-pred1(X,Y),type(Y,const1),
       pred2(X,Z),type(Z,const2).
6  Use predicates and constants that appear
       in the given sentence.
7  Capitalize variables: X, Y, Z, etc.
```

DeiSAM uses a specific rule format that describes the relations of objects and attributes. For example, a fact `on(person,boat)` in a scene graph is decomposed into multiple facts `on(obj1,obj2)`,

| Method | Mean Average Precision (%) ↑ | |
| | DeiVG$_2$ | DeiVG$_1$ |
| --- | --- | --- |
| GroundedSAM | 19.27 | 8.82 |
| DeiSAM (ours) | **81.42** | **78.18** |

Table 1: **DeiSAM handles deictic prompting.** Mean Average Precision (mAP) of DeiSAM and GroundedSAM on Deictic VG datasets are shown. Subscript numbers indicated the complexity (hops on scene graph) of prompts.

`type(obj1,person)`, and `type(obj2,boat)` to account for several entities with the same name in the scene.

## Reasoning with Deictic Prompting

We build a reasoning function $f_{reason} : \mathcal{G} \times \mathcal{R} \to \mathcal{T}$ where $\mathcal{G}$ is a set of facts that represent a scene graph, $\mathcal{R}$ is a set of rules generated by an LLM, and $\mathcal{T}$ is a set of facts that represent identified target objects in the scene.

**(Differentiable) Forward Reasoning.** For a given set $\mathcal{G}$, a *valuation vector* $\mathbf{v} \in [0,1]^{|\mathcal{G}|}$ maps each fact to a corresponding confidence score. DeiSAM incorporates graph neural networks, which pass messages on reasoning graphs that represent a set of rules and update valuation vectors, inferring new facts (Shindo et al. 2023b). To complete object segmentation, DeiSAM identifies target objects as facts, *e.g.* `target(obj1)`, and subsequently extracts the bounding box of the targets from the scene graph. We provide more details in the appendix.

**Semantic Unifier.** DeiSAM unifies diverging semantics in the generated rules and scene graph using concept embeddings similar to neural theorem provers (Rocktäschel and Riedel 2017). We rewrite the corresponding rules $\mathcal{R}$ of a prompt by identifying the most similar terms in the scene graph for each predicate and constant. If rule $R \in \mathcal{R}$ contains a term $x$, which does not appear in scene graph $\mathcal{G}$, we compute the similarity score by

$$\underset{y \in \mathcal{G}}{\arg \max} \, encoder(x)^\top \cdot encoder(y), \quad (1)$$

where $encoder$ is an embedding model for texts.

## Experiments

With the methodology of DeiSAM established, we subsequently provide empirical and qualitative evidence of its benefits over purely neural approaches.

## Experimental Setup

To assess deictic object segmentation, we propose a novel benchmark, the Deictic Visual Genome (DeiVG), which is an extension of the Visual Genome dataset (Krishna et al. 2017). DeiVG consists of visual scenes paired with deictic prompts targeting one or multiple objects in the image. We automatically synthesize prompts from scene graphs in Visual Genome using textual templates. For example, the relations `on(cable,table)` and `behind(cable,mug)`, would yield a prompt *"An object on the table and behind the mug."* targeting the cable. Entries in the DeiVG dataset

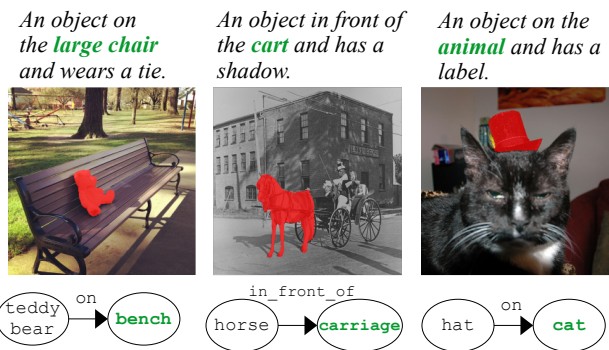

*An object on the **large chair** and wears a tie.*

*An object in front of the **cart** and has a shadow.*

*An object on the **animal** and has a label.*

Figure 3: **DeiSAM can reason on ambiguous prompts.** Segmentation results (middle) on prompts (top) that contain entities not appearing in the scene graphs (bottom). DeiSAM successfully identified objects given two different semantics.

can be categorized by the number of relations they use in their object description. Overall, we generate 3k pairs of a visual scene and a deictic prompt with one relation and 10k pairs of them with two relations that we denote as DeiVG$_1$ and DeiVG$_2$, respectively.

As an evaluation metric, we use mean average precision (mAP) over object classes. Since the object segmentation quality largely depends on the used segmentation model, we focus on assessing the object identification preceding the segmentation step. The DeiSAM configuration for the subsequent experiments uses the ground truth scene graphs from the Visual Genome (Krishna et al. 2017), `gpt-3.5-turbo`[2] as LLM for rule generation, `ada-002`[3] as embedding model for semantic unification, and SAM for object segmentation. Additionally, we provide few-shot examples of deictic prompts and paired rules in the input context of the LLM, which improves performance.

## Empirical Evidence

We compare DeiSAM on DeiVG datasets with GroundedSAM, which combines SAM (Kirillov et al. 2023) with Grounding DINO (Liu et al. 2023b). We report the scores for both methods in Tab. 1. DeiSAM outperforms the purely neural approach by a large margin on both DeiVG$_1$ and DeiVG$_2$. Interestingly, both methods achieve better scores for the seemingly more complex task, a phenomenon that we explore in more detail in the next section.

## Qualitative Evaluation

After empirically demonstrating DeiSAM's capabilities, we look into some qualitative examples. In Fig. 3, we demonstrate the efficacy of the semantic unifier. All examples use terminology in the deictic prompt diverging from the scene graph entity names. Nonetheless, the unification step successfully maps synonymous terms and still produces the cor-

---

[2]https://openai.com/blog/introducing-chatgpt-and-whisper-apis

[3]https://openai.com/blog/new-and-improved-embedding-model

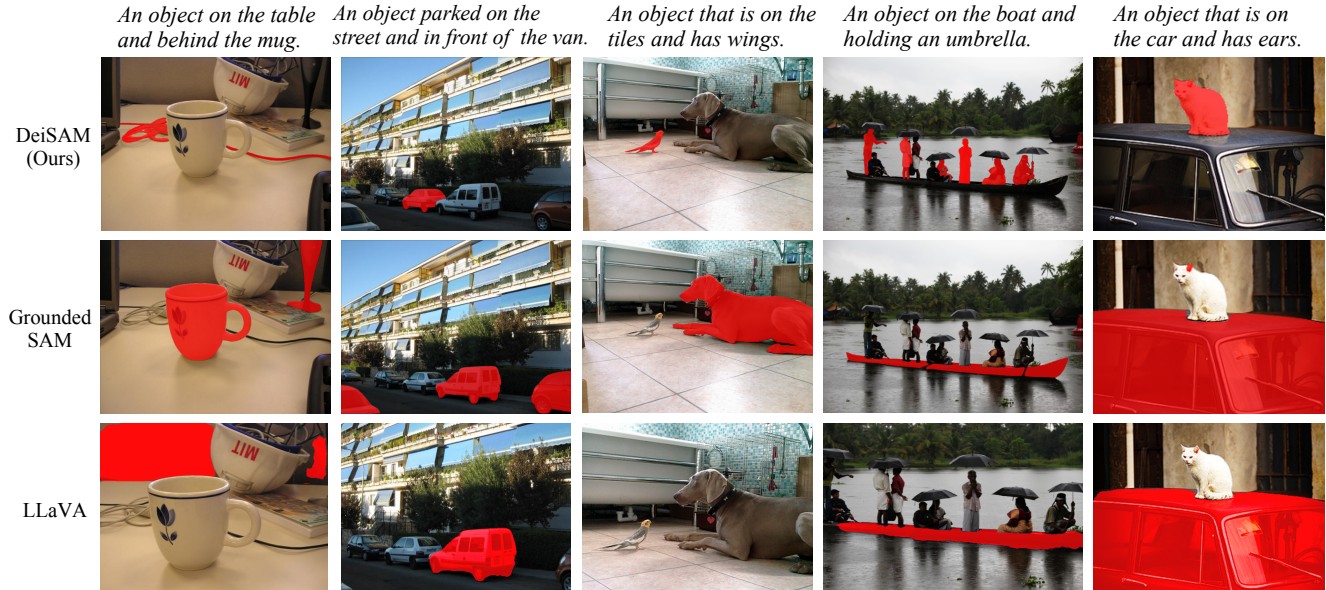

Figure 4: **DeiSAM segments objects with deictic prompts.** Segmentation results on the DeiVG dataset using DeiSAM, GroundedSAM, and LLaVA are shown with deictic prompts on the top. DeiSAM correctly identifies and segments objects given deictic prompts (top row), while GroundedSAM and LLaVA often segment a wrong object or fail to identify an object (bottom rows). More results are available in Fig. 6 in the Appendix. (Best viewed in color)

rect segmentation masks, overcoming the limitation of off-the-shelf symbolic logic reasoners.

In Fig. 4, we further compare DeiSAM with Grounded-SAM and interactive LLaVA (Liu et al. 2023a; Chen et al. 2023). DeiSAM produces the correct segmentation mask even for complicated shapes (*e.g.* partially occluded cable) or complex scenarios (*e.g.* multiple people, only some holding umbrellas). GroundedSAM and LLaVA, however, regularly fail to identify the correct object. More results are available in the Appendix. Overall, the examples further highlight DeiSAM's capability of complex reasoning for object segmentation, outperforming pure neural approaches.

## Discussion

In our experiments, we observed degraded performance on $DeiVG_1$ compared to $DeiVG_2$ for both models. This result may seem counterintuitive since $DeiVG_1$ contains the simpler prompts. We attribute the gap to inconsistencies in the original Visual Genome dataset itself. For example, only one of multiple objects in an image might be labeled correctly in the scene graph. Consequently, our derived DeiVG benchmark can contain prompts with ambiguous target objects. Simple prompts are generally more ambiguous (e.g., *"an object on the table"*) thus, they are disproportionally affected by this issue. For future work, we aim to improve the DeiVG benchmark by cleaning up inconsistent prompts.

Moreover, we observed that neural baselines are easily confounded by objects mentioned in the prompt, *e.g.* given *"An object that is on the car"*, the car itself is segmented, discarding the intended relation. In contrast, DeiSAM successfully segments objects given prompts requiring rela-

tional reasoning since it embraces logic reasoners and encodes relations of objects explicitly.

While DeiSAM achieves impressive results, it is worth considering some of the limitations of this work. Our current experimental setup uses ground-truth scene graphs from Visual Genome, whereas the errors of an actual scene graph generator may result in a worse performance. However, DeiSAM's modularity accommodates recent advances in scene graph generations, *e.g.* unbiased scene graph generators (Sudhakaran et al. 2023).

Finally, DeiSAM may be leveraged for gradient-based learning approaches, since the reasoning is differentiable. For future work, we plan to pass gradients through DeiSAM to the scene generator and LLM submodules. Such a setup would allow for fine-tuning neural modules to generate high-quality scene graphs and logic rules with reasoning explicitly modeled in the training pipeline. Additionally, our setup allows for structure learning of logic rules from segmentation examples, which is a promising research direction.

## Conclusion

We proposed DeiSAM to perform deictic image segmentation. DeiSAM embraces large-scale neural networks to understand complex prompts with visual scenes and performs differentiable forward reasoning to identify objects. DeiSAM allows users to describe a target using relations of objects flexibly. Moreover, we proposed the novel Deictic Visual Genome (DeiVG) benchmark for segmentation with complex deictic prompts. In our extensive experiments, we demonstrated that DeiSAM outperforms neural baselines highlighting its strong reasoning capabilities on visual scenes with complex textual prompts.

## Acknowledgements

This work was supported by the Federal Ministry of Education and Research (BMBF) AI lighthouse project "SPAICER" (01MK20015E), the EU ICT-48 Network of AI Research Excellence Center "TAILOR" (EU Horizon 2020, GA No 952215), and the Collaboration Lab "AI in Construction" (AICO). The work has also benefted from the Hessian Ministry of Higher Education, Research, Science and the Arts (HMWK) cluster projects "The Third Wave of AI" and "The Adaptive Mind". We gratefully acknowledge support by the German Center for Artificial Intelligence (DFKI) project "SAINT". It also benefited the HMWK / BMBF ATHENE project "AVSV".

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

## Details of Forward Reasoning

DeiSAM employs a graph neural network-based differentiable forward reasoner (Shindo et al. 2023b), and we briefly explain the reasoning process. We represent a set of (weighted) rules as a bipartite graph as shown in Fig. 5.

**Definition 1.** *A Forward Reasoning Graph is a bipartite directed graph* $(\mathcal{V}_{\mathcal{G}}, \mathcal{V}_{\wedge}, \mathcal{E}_{\mathcal{G} \to \wedge}, \mathcal{E}_{\wedge \to \mathcal{G}})$, *where* $\mathcal{V}_{\mathcal{G}}$ *is a set of nodes representing ground atoms (atom nodes),* $\mathcal{V}_{\wedge}$ *is set of nodes representing conjunctions (conjunction nodes),* $\mathcal{E}_{\mathcal{G} \to \wedge}$ *is set of edges from atom to conjunction nodes and* $\mathcal{E}_{\wedge \to \mathcal{G}}$ *is a set of edges from conjunction to atom nodes.*

DeiSAM performs forward-chaining reasoning by passing messages on the reasoning graph. Essentially, forward reasoning consists of *two* steps: (1) computing conjunctions of body atoms for each rule and (2) computing disjunctions for head atoms deduced by different rules. These two steps can be efficiently computed on bi-directional message-passing on the forward reasoning graph.

**(Direction $\to$) From Atom to Conjunction.** First, messages are passed to the conjunction nodes from atom nodes. For conjunction node $v_i \in \mathcal{V}_{\wedge}$, the node features are updated:

$$v_i^{(t+1)} = \bigvee \left( v_i^{(t)}, \bigwedge_{j \in \mathcal{N}(i)} v_j^{(t)} \right), \quad (2)$$

where $\bigwedge$ is a soft implementation of *conjunction*, and $\bigvee$ is a soft implementation of *disjunction*. Intuitively, probabilistic truth values for bodies of all ground rules are computed softly by Eq. 2.

**(Direction $\leftarrow$) From Conjunction to Atom.** Following the first message passing, the atom nodes are then updated using the messages from conjunction nodes. For atom node $v_i \in \mathcal{V}_{\mathcal{G}}$, the node features are updated:

$$v_i^{(t+1)} = \bigvee \left( v_i^{(t)}, \bigvee_{j \in \mathcal{N}(i)} w_{ji} \cdot v_j^{(t)} \right), \quad (3)$$

where $w_{ji}$ is a weight of edge $e_{j \to i}$. We assume that each rule $C_k \in \mathcal{C}$ has its weight $\theta_k$, and $w_{ji} = \theta_k$ if edge $e_{j \to i}$ on the reasoning graph is produced by rule $C_k$. Intuitively, in Eq. 3, new atoms are deduced by gathering values from different ground rules and from the previous step.

We used product for conjunction, and *log-sum-exp* function for disjunction:

$$softor^{\gamma}(x_1, \ldots, x_n) = \gamma \log \sum_{1 \le i \le n} \exp(x_i / \gamma), \quad (4)$$

where $\gamma > 0$ is a smooth parameter. Eq. 4 approximates the maximum value given input $x_1, \ldots, x_n$.

**Prediction.** The probabilistic logical entailment is computed by the bi-directional message-passing. Let $\mathbf{x}_{atoms}^{(0)} \in [0,1]^{|\mathcal{G}|}$ be input node features, which map a fact to a scalar value, $\mathsf{RG}$ be the reasoning graph, $\mathbf{w}$ be the rule weights, $\mathcal{B}$ be background knowledge, and $T \in \mathbb{N}$ be the infer step. For fact $G_i \in \mathcal{G}$, DeiSAM computes the probability as:

$$p(G_i \mid \mathbf{x}_{atoms}^{(0)}, \mathsf{RG}, \mathbf{w}, \mathcal{B}, T) = \mathbf{x}_{atoms}^{(T)}[i], \quad (5)$$

where $\mathbf{x}_{atoms}^{(T)} \in [0,1]^{|\mathcal{G}|}$ is the node features of atom nodes after $T$-steps of the bi-directional message-passing.

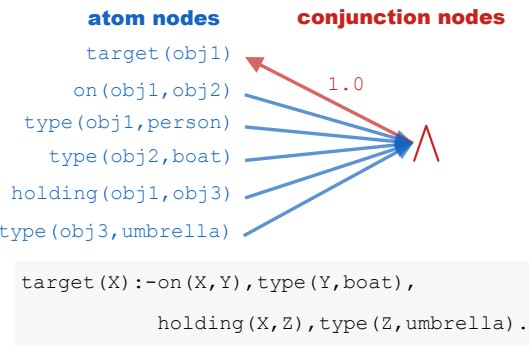

Figure 5: A partial reasoning graph (top) and a corresponding rule (bottom). Blue nodes represent facts, red nodes represent conjunctions, and edges represent a ground rule. DeiSAM performs differentiable forward reasoning by performing bi-directional message passing on the reasoning graph using soft-logic operations to aggregate messages. Only relevant nodes are shown. (Best viewed in color)

## Details of Experiments

We provide details of the models used in the evaluation.

**DeiSAM.** We used NEUMANN (Shindo et al. 2023b) with $\gamma = 0.01$ for soft-logic operations, and the number of inference steps is set to 2. We used a publicly available checkpoint[4] for the SAM model. We set the box threshold to 0.3 and the text threshold to 0.25 for the SAM model. All generated rules are assigned a weight of 1.0.

For LLMs, we provided few-shot examples of deictic prompts and desired outputs in the input context, *e.g.*

```
1  Example:
2  An object that is next to the cup.
3  available predicates: next_to
4  target(X):-next_to(X,Y),type(Y,cup).
```

These few-shot examples improved the quality of the rule generation that follows a certain format.

**GroundedSAM.** We used a pre-trained and publicly available checkpoint[4] for the SAM model and a public checkpoint[5] for Grounding DINO. We set the box threshold to 0.3 and the text threshold to 0.25 for the SAM model.

**Evaluation Metric.** We used mean average precision (mAP) to evaluate segmentation models. Segmentation masks are converted to corresponding bounding boxes by computing their contours, and then mAP is computed by comparing them with the ground truth bounding boxes provided by Visual Genome.

## Additional Segmentation Results

We provide supplementary results of the segmentation on the DeiVG dataset in Fig. 6.

---

[4]https://huggingface.co/spaces/abhishek/StableSAM/blob/main/sam_vit_h_4b8939.pth

[5]https://github.com/IDEA-Research/GroundingDINO/releases/download/v0.1.0-alpha2/groundingdino_swinb_cogcoor.pth

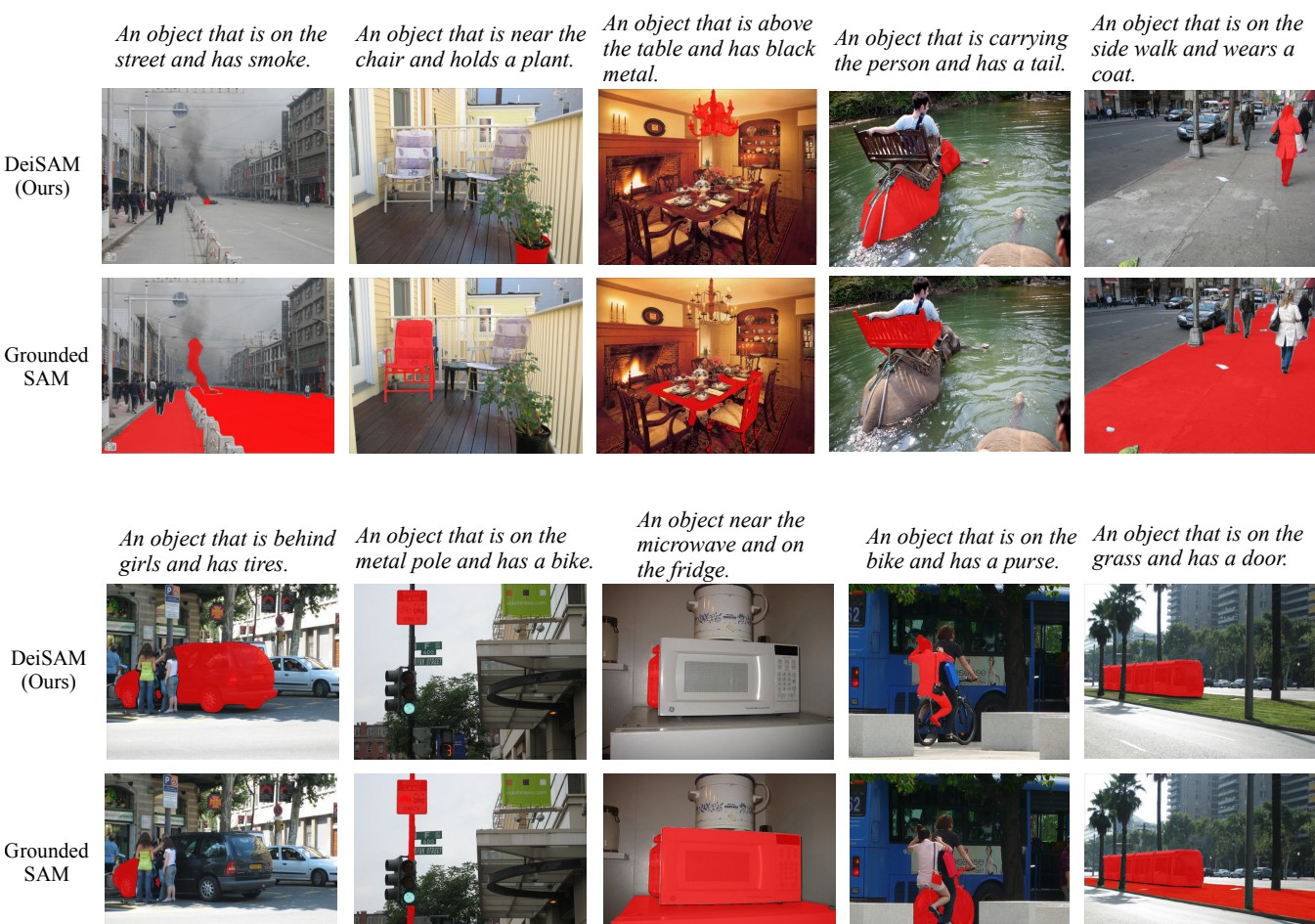

Figure 6: Segmentation results on the DeiVG dataset using DeiSAM and GroundedSAM are shown with deictic prompts.