# OpenReview forum: "DeiSAM: Segment Anything with Deictic Prompting"
_AAAI.org/2024/Workshop/NuCLeaR — NuCLeaR 2024_

### Official Review · Reviewer_UKfE · 2023-12-07
**Interesting work!**

**Rating:** 7
**Confidence:** 3

**Review:**

The paper proposes a framework to perform complex segmentation tasks when instructed in language queries. Direct prompting or direct inference for several segmation models suffer under complex language instruction where the model has to perform grounded reasoning and identify relative attributes of objects with each other.
Deictic prompting proposes to use LLMs for generating reasoning programs which can then be grounded in the image by using scene graphs. With this the segmentation model can focus only on target areas.
Empirically the authors observe dramatically high improvement in performance as compared to baselines.

I have one weakness which is to improve readability of the paper by adding more examples and description of forward reasoning.

---

### Official Review · Reviewer_Y69e · 2023-12-07
**The paper introduces a novel concept for solving reasoning-based segmentation. Overall this is a relevant paper for the conference.**

**Rating:** 8
**Confidence:** 4

**Review:**

Strengths:

1. Authors propose a novel concept of reasoning with deictic prompting that is very relevant for reasoning-based image segmentation, a known drawback of captioning models.
2. Embedding based unification allows for use of individual entity information.
3. The result indicates that the technique works.

Weakness:
1. Lack of comparison with existing datasets.

Formatting:
1. Some of the appendix details (eg. deictic prompt sample) should be moved to the main paper.

Questions:
1. Did the authors try to directly rewrite the query using an LLM to extract the relevant components from natural language intent (eg. InstructBLIP, LLaVaInstruct)

---

### Official Review · Reviewer_oCVP · 2023-12-08
**Interesting Paper on LLM+FOL for segmentation tasks**

**Rating:** 7
**Confidence:** 3

**Review:**

The paper proposes DeiSAM, a neuro-symbolic pipeline for segmenting objects in images based on deictic prompts, which are natural language descriptions that refer to objects depending on the context. The paper also introduces a novel benchmark, DeiVG, which contains visual scenes paired with complex deictic prompts. The paper shows that DeiSAM outperforms neural baselines on the deictic segmentation task and demonstrates its reasoning capabilities.

### **Strength**

- Using LLM to generate first-order logic is a very promising direction for the neuro-symbolic approach, which combines the power of neural representation into the commonsense reasoning in LLM. Similar approaches have been proposed in [1].
- The scope is clearly stated and empirical results against grounded SAM are impressive.

[1] Hsu, Joy, et al. "What's Left? Concept Grounding with Logic-Enhanced Foundation Models." arXiv preprint arXiv:2310.16035 (2023).

### **Weakness**

- More details on LLaVa baselines will be helpful, as it belongs to another category. How do you compare the visual instruction tuning baseline with prompting?

---

### Decision · Program_Chairs · 2023-12-11

Accept